

# GPX2 predicts recurrence-free survival and triggers the Wnt/β-catenin/EMT pathway in prostate cancer

Ming Yang[1], Xudong Zhu[1], Yang Shen[1], Qi He[1], Yuan Qin[1], Yiqun Shao[2], Lin Yuan[3] and Hesong Ye[1]

[1] The Second Affiliated Hospital of Nanjing University of Chinese Medicine, Nanjing, China
[2] Yueyang Hospital of Integrated Traditional Chinese and Western Medicine Affiliated to Shanghai University of Traditional Chinese Medicine, Shanghai, China
[3] Affiliated Hospital of Nanjing University of Chinese Medicine, Nanjing, China

## ABSTRACT

**Objective**. This study aimed to establish a prognostic model related to prostate cancer (PCa) recurrence-free survival (RFS) and identify biomarkers.

**Methods**. The RFS prognostic model and key genes associated with PCa were established using Least Absolute Shrinkage and Selection Operator (LASSO) and Cox regression from the Cancer Genome Atlas (TCGA)-PRAD and the Gene Expression Omnibus (GEO) GSE46602 datasets. The weighted gene co-expression network (WGCNA) was used to analyze the obtained key modules and genes, and gene set enrichment analysis (GSEA) was performed. The phenotype and mechanism were verified in vitro.

**Results**. A total of 18 genes were obtained by LASSO regression, and an RFS model was established and verified (TCGA, AUC: 0.774; GSE70768, AUC: 0.759). Three key genes were obtained using multivariate Cox regression. WGCNA analysis obtained the blue module closely related to the Gleason score ($cor = -0.22$, $P = 3.3e-05$) and the unique gene glutathione peroxidase 2 (GPX2). Immunohistochemical analysis showed that the expression of GPX2 was significantly higher in patients with PCa than in patients with benign prostatic hyperplasia ($P < 0.05$), but there was no significant correlation with the Gleason score (GSE46602 and GSE6919 verified), which was also verified in the GSE46602 and GSE6919 datasets. The GSEA results showed that GPX2 expression was mainly related to the epithelial–mesenchymal transition (EMT) and Wnt pathways. Additionally, GPX2 expression significantly correlated with eight kinds of immune cells. In human PCa cell lines LNCaP and 22RV1, si-GPX2 inhibited proliferation and invasion, and induced apoptosis when compared with si-NC. The protein expression of Wnt3a, glycogen synthase kinase 3β (GSK3β), phosphorylated (p)-GSK3β, β-catenin, p-β-catenin, c-myc, cyclin D1, and vimentin decreased; the expression of E-cadherin increased; and the results for over-GPX2 were opposite to those for over-NC. The protein expression of GPX2 decreased, and β-catenin was unchanged in the si-GPX2+ SKL2001 group compared with the si-NC group.

**Conclusion**. We successfully constructed the PCa RFS prognostic model, obtained RFS-related biomarker GPX2, and found that GPX2 regulated PCa progression and triggered Wnt/β-catenin/EMT pathway molecular changes.

Corresponding authors
Lin Yuan, yuanlin47@163.com
Hesong Ye, yehesong@163.com

## INTRODUCTION

Prostate cancer (PCa) is one of the most common male malignant tumors in the United States and the second leading cause of cancer-related deaths in men (*Kang et al., 2020*). More than 80% of PCa cases are diagnosed as local diseases and usually treated by radical prostatectomy. However, about 15% of patients have a biochemical recurrence within 5 years after surgery, and the recurrence rate has been reported to be as high as 40% within 10 years. Local PCa that relapses after treatment can progress to fatal castration-resistant prostate cancer (CRPC) (*Li et al., 2017*). The causes of PCa recurrence are complex and diverse, and the specific mechanism has not yet been clarified (*Siegel, Miller & Jemal, 2019*). Therefore, research on the mechanism of PCa recurrence and the application of prognostic biomarkers may be of great significance in improving the survival rate of patients with PCa.

Many studies have shown that the epithelial–mesenchymal transition (EMT) and Wnt/β-catenin signaling pathways play an essential role in the occurrence and development of PCa (*Montanari et al., 2017*). EMT is necessary for PCa occurrence and distant metastasis, and plays a critical role in PCa metastasis to other organs (*He et al., 2020*). Epithelial cells attain the biological characteristics of stromal cells (*Chaves et al., 2021*). Studies have shown that the EMT and Wnt/β-catenin signaling pathways are closely related. Wnt binding to its receptor frizzled protein results in protein phosphorylation, which inhibits GSK-3 β activity. Consequently, β-catenin degradation is blocked and β-catenin accumulates in the cytoplasm, enters the nucleus, interacts with cytokines, activates the transcription of downstream target genes, induces EMT in cells, and promotes tumor growth and metastasis (*Hseu et al., 2019*; *Sun et al., 2020*).

Bioinformatics analysis is one of the crucial methods used for gene molecular research based on Big Data (*Hutter & Zenklusen, 2018*; *Botía et al., 2017*). In this study, PCa RFS–related differentially expressed genes (DEGs) were screened by analyzing the data of PCa-related gene expression and clinicopathological characteristics in the Cancer Genome Atlas (TCGA) and Gene Expression Omnibus (GEO) databases. We analyzed the protein–protein interaction (PPI) based on DEGs. Survival, Cox regression, and LASSO regression analyses were used to establish and verify the prognostic model. The DEGs between different Gleason scores of PCa tissues and the key gene glutathione peroxidase 2 (GPX2) were obtained by weighted gene co-expression network analysis (WGCNA). The GSEA of GPX2 and its significance in prognosis and immunity were analyzed. Finally, a series of *in vitro* experiments were conducted to explore the potential role of GPX2 in PCa, so as to provide new clues for diagnosing and treating PCa.

## MATERIALS AND METHODS

### Data acquisition

The data of gene expression profiles from the TCGA-PRAD dataset were downloaded from the TCGA database and standardized. At the same time, the clinical information of patients, including age, gender, TNM stage, pathological stage, and prognosis, was downloaded. The samples with incomplete clinical information and survival data were excluded. A total of 481 PCa samples and 51 adjacent tissue samples were included in the study. Three datasets of gene expression and clinical profiles of Pca were downloaded from the GEO database (GSE70768, GSE46602, and GSE6919).

### Construction and validation of the Pca RFS prognostic model

The data were analyzed using the R software DESeq and Limma package. The volcano and heat maps were drawn using the $P$ value <0.05 and |logfc|>2 as the screening conditions. After taking the intersection, Pca DEGs were obtained. The PPI network of DEGs was constructed using the STRING database, and the setting was adjusted to the interactive score of 0.7. Cytohubba and MCODE modules were used to screen Top30 and topology-related genes from Cytoscape 3.7.2. The prognostic model was established using univariate Cox, LASSO, and multivariate Cox regression analyses. Finally, the model was validated in the TCGA-PRAD and GSE70768 datasets.

### WGCNA and GPX2 predicted PCa RFS

A total of 2,191 TCGA DEGs were analyzed using WGCNA. The Pearson correlation coefficient between genes was calculated. The scale-free network was constructed and the appropriate threshold was selected for network construction. Using two-step construction, the adjacency matrix was transformed into a topological overlap matrix, the clustering tree was generated through hierarchical clustering, and clustering was combined through a dynamic cut. The significance of gene and module was estimated, and the clinical sample grouping information was obtained. The identity of each gene module was calculated to measure the importance of genes in each module. Setting parameters |gene module|>0.8 and |gene significance|>0.2 as criteria, we screened the hub genes of modules closely related to clinical traits. The blue module was found to be significantly related to the Gleason score. The key genes and blue modules were crossed, and the single gene GPX2 was obtained. The TCGA-PRAD dataset verified the GPX2-predicted PCa RFS.

### GSEA and immunohistochemical (IHC) analysis

GSEA 4.0.1 was used to compare and analyze the DEGs between high- and low-expression groups of GPX2 in the PCa tumor samples of the TCGA-PRAD dataset. Gene set database selection KEGG v7. 4 was used to set the replacement times to 1,000; $P < 0.05$ and false discovery rate (FDR) <0.25 indicated significantly enriched genes. Between May 2021 and January 2022, the Second Affiliated Hospital of Nanjing University of Chinese Medicine collected tissues from 20 patients with PCa (10 patients with a high Gleason score $\geq$8 and 10 patients with a low Gleason score $\leq$7) and 10 patients with benign prostatic hyperplasia. Detailed information is shown in Table S1. All patients signed an informed

consent form. This study was approved by the ethics committee of the Second Affiliated Hospital of the Nanjing University of Chinese Medicine (2021SEZ-030-01). The biopsy samples were collected, fixed with 10% formaldehyde, and embedded in paraffin after routine treatment. The prepared wax block was cut into sections with a thickness of 2 μm. Immunohistochemical staining was performed on the treated sections. The processed sections were stained with GPX2 (ab140130; Abcam, Cambridge, UK). Two pathologists used the double-blind method to judge each slice. The sections were observed using a low-power mirror under a microscope to select the best field of vision, and then a high-power lens was used in this range $10 \times 40$. Five visual fields were randomly observed, and the IHC score was defined as the product of the frequency of positive cells and the intensity of staining.

## GPX2 expression with immune cells in PCa

In this study, the CIBERSORT algorithm was used to calculate the infiltration proportion of 22 kinds of immune cells in PCa tissue, and 481 PCa samples were analyzed using the R software. The samples with a $P$ value <0.05 were included in the follow-up analysis. Taking the median expression level of GPX2 mRNA as the boundary, the samples were divided into high- and low-expression groups.

## Cell culture and qRT-PCR analysis

Human PCa cell lines (PC-3, DU145, LNCaP, and 22RV1) were purchased from Procell (Wuhan, China). The cells were cultured in RPMI-1640 (bl303a; Biosharp, Anhui, China) at 37 °C and in the presence of 5% CO2. The cells adhered to the wall and were passaged every 3 days. The cells in the logarithmic growth phase were used for the experiment. TRIzol reagent (bs259a, Biosharp, China) was used to extract the total RNA of PC-3, LNCaP, 22Rv1, and DU145 cells. A reverse transcription kit (11119es60; Yeasen, Shanghai, China) was used to reverse transcribe RNA into cDNA, and a SYBR Green Kit (11201es50; Yeasen, China) was used for qRT-PCR amplification, with β -actin as an internal control. The 2- Δ ΔCT method was used for calculation. The primers were as follows: GPX2: 5′-GCCTCCTTAAAGTTGCCATA-3′ and 5′-GCCCAGAGCTTACCCA-3′; β -actin: 5′-GAAGAGA-GAGACCCTCACGCTG-3′, and 5′-ACTGTGAGGAGGGGAGATTCAGT-3′. The experiment was repeated three times.

## Transfection and grouping

LNCaP and 22RV1 cells in the logarithmic growth phase ($n = 200,000$) were inoculated into the cell culture plate and transfected according to the Lipofectamine 2000 (11668-027, Invitrogen, Waltham, MA, USA) instructions. They were divided into GPX2 low expression (si-GPX2) and negative control (si-NC) groups, GPX2 overexpression (over-GPX2) and negative control (over-NC) groups, and SKL2001(HY-101085, MCE, USA) +si-GPX2 groups. The transfection effect was verified using qRT-PCR and Western blotting. The experiment was repeated three times.

## CCK-8 assay

LNCaP and 22RV1 cells in the logarithmic growth phase ($n = 2,000$) were inoculated into the cell culture plate. After undergoing corresponding treatment according to experimental

grouping, 10 μL of cells were added to each well containing CCK-8 solution (PR645; Dojindo, Kumamoto, Japan). The culture plate was incubated and the absorbance value was determined to be 450 nm using a microplate reader (SpectraMax i3; Molecular Devices, San Jose, CA, USA). Cell proliferation inhibition rate = (control group absorbance value– experimental group absorbance value)/control group absorbance value ×100%. The experiment was repeated three times.

### Flow cytometry assay

The transfected cells were collected and digested with trypsin without EDTA. The adherent cells were collected and centrifuged. The supernatant was discarded and the cell precipitate was washed twice with phosphate-buffered saline (PBS). Annexin V–FITC/PI (556547, BD; Franklin Lakes, NJ, USA) was added. After incubation in the dark at room temperature for 5 min, we detected the apoptosis rate of LNCaP and 22RV1 cells using a flow cytometer (LSRII instrument; BD, Franklin Lakes, NJ, USA). The experiment was repeated three times.

### Transwell invasion assay

The transfected cells were collected and the cell concentration was adjusted to $3\times 10^5$/mL. The cells were inoculated into the upper layer of the Transwell chamber (3422; Corning, Corning, NY, USA) which contained a serum-free medium, and 100 μL/well of the cell suspension was added. Additionally, 600 μL of the fresh culture medium was added to the lower layer of the chamber. The liquid in the upper chamber was discarded after culturing for 24 h, and the cells were wiped off the upper-chamber membrane with a wet cotton swab. The cells on the lower-chamber membrane were fixed with methanol for 20 min, dyed with crystal violet, rinsed with PBS until the background was clean, dried, and imaged after sealing. ImageJ software was used to count the number of transmembrane cells. The experiment was repeated three times.

### Western blot analysis

The total protein was extracted from human PCa cells following the instructions of the total protein extraction kit (bl521a; Biosharp, Shandong, China); subsequently, the protein concentration was detected using the diquinoline formic acid method. The denatured protein samples were separated by electrophoresis according to which membrane was transferred using the semi-dry method. After sealing the membrane with skimmed milk powder for 2 h, we added β-actin (gb12001; Servicebio, Wuhan, China), Wnt3a (2721; Cell Signaling Technology, Danvers, MA, USA), GSK3 β (ab2602; Abcam, UK), phosphorylated (p)-GSK3 β Ser9 (ab131097; Abcam, UK), β-catenin (ab32572; Abcam, UK), p- β-catenin (ab27798; Abcam, UK), C-myc (ab32072; Abcam, UK), Cyclin D1 (2978; Cell Signaling Technology, USA), vimentin (3195; Cell Signaling Technology, USA), E-cadherin (60330-I-Ig, Proteintech, USA), and GPX2 (ab140130; Abcam, UK) antibodies. Subsequently, the membrane was incubated overnight at 4 °C, then incubated with primary and secondary antibodies at room temperature for 2 h, exposed, and developed using the ECL film. The protein expression was analyzed, and the experiment was repeated three times.

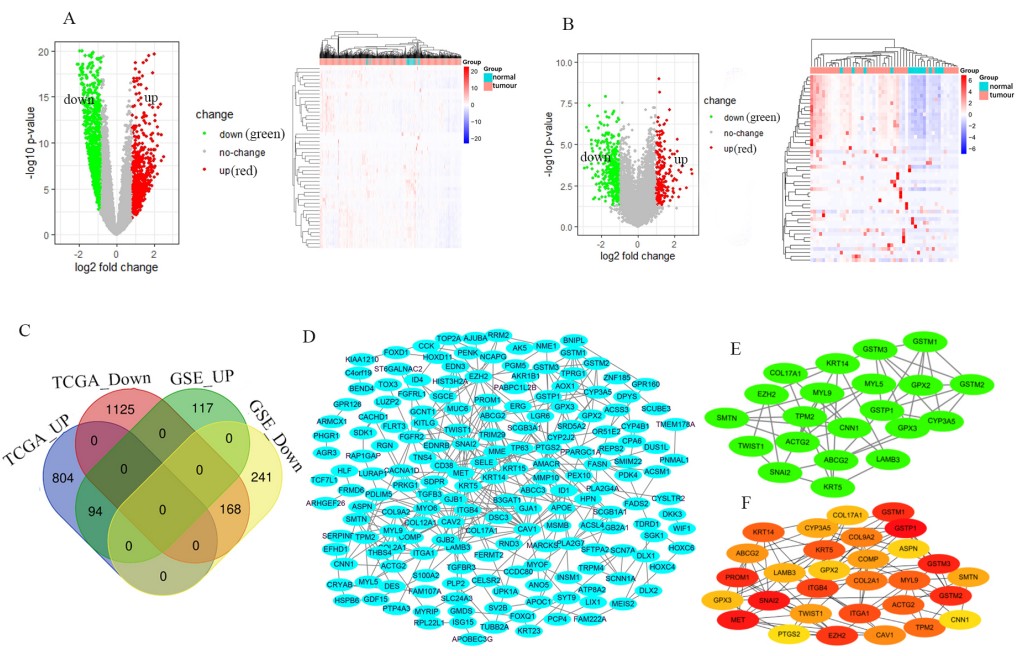

**Figure 1 Identification of the DEGs.** Volcano and heat maps of two datasets TCGA-PRAD (A) and GSE46602 (B). Red nodes represent upregulated DEGs, the green node indicates downregulated DEGs. (C) Venn diagrams of DEGs from the aforementioned two datasets. (D) Using the STRING online database and Csytoscape 3.7.2 to construct the PPI network of the DEGs. The Top30 and topology module genes were screened using Cytohubba and MCODE app in the Cytoscape software.

## Statistical analysis

The Student $t$-test was used for continuous variables, while the classification variables were analyzed using the $\chi^2$ test. Cox and LASSO regression models were used to analyze the predictors of RFS. The data were expressed as mean $\pm$ standard deviation. All data were analyzed with R version 4.1.2, SPSS 24.0 and GraphPad Prism 8.0. A $P$ value <0.05 indicated a significant difference. All tests were repeated three times.

## RESULTS

### Identified DEGs

The GSE46602 dataset had 211 upregulated genes and 409 downregulated genes. The TCGA-PRAD dataset was comprised of 898 upregulated genes and 1,293 downregulated genes. The volcano and heat maps showed DEGs (Figs. 1A and 1B). After further taking the intersection of the aforementioned datasets, we obtained the common 262 DEGs (94 upregulated genes and 168 downregulated genes) (Table 1 and Fig. 1C). STRING and Cytoscape were used to construct the PPI network of DEGs (Fig. 1D). Cytohubba and MCODE modules were used to screen Top30 and topology-related DEGs (Figs. 1E and 1F).

**Table 1  A total of 262 DEGs were identified from the TCGA-PRAD and GSE46602 datasets, with 94 upregulated and 168 downregulated.**

| DEGs | Gene names |
| --- | --- |
| Upregulated | GDF15 ERG B3GAT1 COL9A2 DLX2 APOE GJB1 TOX3 STX19 PKIB NAALADL2 ARHGEF26 REPS2 LUZP2 PTP4A3 TMEM178A GPR160 BEND4 ELL3 ASPN CGREF1 PHGR1 NEK5 INSM1 PPP1R14B THBS4 HOXC4 KCNG3 COMP SMIM31 PLA2G7 SBK1 OR51E2 MARCKSL1 AK5 SH3RF1 DUS1L ATP8A2 BEND3 PABPC1L2B MMP10 NME1 TWIST1 FAM222A APOC1 COL12A1 PDLIM5 HPN AGR3 VSTM2L FASN H2AW SMIM22 MYO6 EZH2 RAP1GAP PODXL2 RPL22L1 HLA-DMB PCDHB2 MS4A8 AMACR RAB17 TRPM4 ISG15 FGFRL1 GLYATL1 CACNA1D SRARP TUBB2A SDK1 ACSM1 SLC43A1 COL2A1 RRM2 TOP2A GJB2 MYL5 SFTPA2 PEX10 MYRIP TMTC4 FOXD1 GMDS HOXC6 TDRD1 DLX1 POPDC3 CYP2J2 NCAPG GCNT1 CRACR2B DNAH5 ERVH48-1 |
| Downregulated | PNCK SLC2A5 HOXD11 RGN UPK1A MET FXYD6 ANP32E KIAA1210 CNN1 PALLD ANO5 DNAJC15 SCUBE3 VWA5A CD38 NEFH MSMB MEIS2 BCL2 C12orf75 TRIM29 ID1 PENK ECRG4 HLF GSTP1 SRD5A2 CPAMD8 NDNF KRT15 LAMB3 PPARGC1A CYP3A5 IER3 SGK1 AOC1 CD177 PTGS2 WIF1 GPX2 MME CAVIN2 CHST2 NR4A3 BNIPL PDK4 LSAMP CXCL17 SERPINB11 ACSS3 TCF7L1 TCEAL2 FAM83B SCGB1A1 GSTM1 DKK3 MUC6 PRIMA1 ARMCX1 AKR1B1 C11orf45 S100A2 MCC WFDC2 TENT5B SMTN SCN7A FLRT3 MYZAP TGFBR3 MYL9 TPM2 CELSR2 AJUBA PGM5 CAV2 NSG1 SLC24A3 GSTM2 AOX1 ACSL4 CYP4B1 HSPB6 MPZL2 LGR6 FOXQ1 GJA1 DEFB1 PLP2 ITGB4 CACHD1 CYSLTR2 CRYAB EFHD1 PCP4L1 ITGA1 PRRG4 FADS2 SELE TMEM252 SV2B C4orf19 MYOF FRMD6 EDN3 PLA2G4A DES ABCC3 FGFR2 AVPI1 EDNRB CAPG FBXO17 DPYS RND3 SCGB3A1 CCK C8orf88 SCGB2A1 LIX1 ARHGAP23 INSYN1 ADGRG6 FERMT2 CPA6 INAVA PRDM8 KRT14 ABCG2 TMEM158 LURAP1 DSC3 TPRG1 TGFB3 AFAP1L2 ID4 PRKG1 PLCL1 SERPINB1 SLC18A2 CCDC80 CAV1 TNS4 PROM1 GSTM3 METTL7A COL17A1 ZNF185 ACTG2 ST6GALNAC2 SCNN1A PCP4 APOBEC3G FAM107A GPX3 PPP1R3C PNMA8A KRT5 SGCE KITLG SLC14A1 NRG1 SYT9 SNAI2 TP63 PARM1 KRT23 |

## PCa RFS prognosis model

A total of 32 genes related to the prognosis of PCa RFS were analyzed using univariate Cox analysis, and a prognostic model based on 18 genes of LASSO regression was constructed: EZH2*0.46 + ELL3*−0.18 + APOC1*−0.04 + NME1*−0.22 + FAM222A*−0.60 + SLC43A1*−0.71 + GCNT1*−0.04 + FOXD1*0.17 + COL2A1*0.028 + GPX2*−0.11 + FOXQ1*0.08 + ID4*−0.32 + IER3*−0.17 + SGCE*−0.13 + ANO5*−0.25 + FBXO17*−0.03 + PNMA8A*0.49 + EDN3*−0.03 (Figs. 2A and 2B). According to the median risk score of the prognostic model, we divided patients with PCa into high-risk

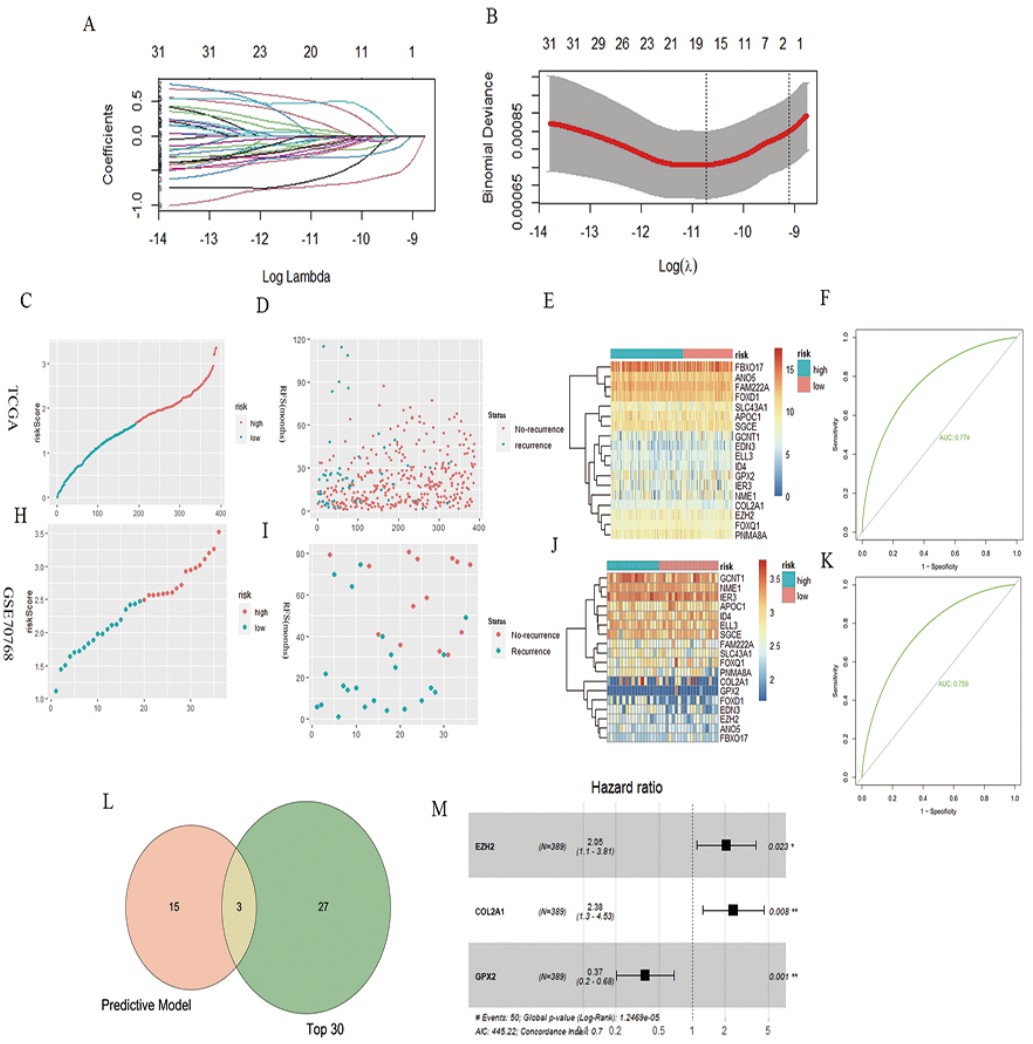

**Figure 2** **PCa RFS predictive model and genes.** (A and B) LASSO coefficient spectrum of 18 RFS predictive model–related genes. (C, D, and E) Risk plot of the RFS predictive model in TCGA. (F) ROC curve in TCGA. (H, I, and J) Risk plot of the RFS predictive model in GSE70768. (F) ROC curve in GSE70768. (L) Venn diagrams of the overlapping DEGs between the predictive model and Top30. (M) Multivariate Cox analysis of three key genes.

and low-risk groups, and the RFS-related scatter plot and the heat map of the prognostic model were constructed (Figs. 2C–2E). The ROC curve of the risk score was also generated, with AUC = 0.774 (Fig. 2F). The GSE70768 dataset was used to verify the prognostic model, and the RFS-related scatter plot and the heat map of the prognostic model were constructed (Figs. 2H–2J); the ROC curve of the risk score had AUC = 0.759 (Fig. 2K). Three key genes obtained by the intersection of the prognostic model and Top30 were GPX2, EZH2, and COL2A1 (Fig. 2L). The multivariate Cox regression showed that the three genes significantly correlated with patient RFS (P = 0.001, 0.023,0.008, Fig. 2M).

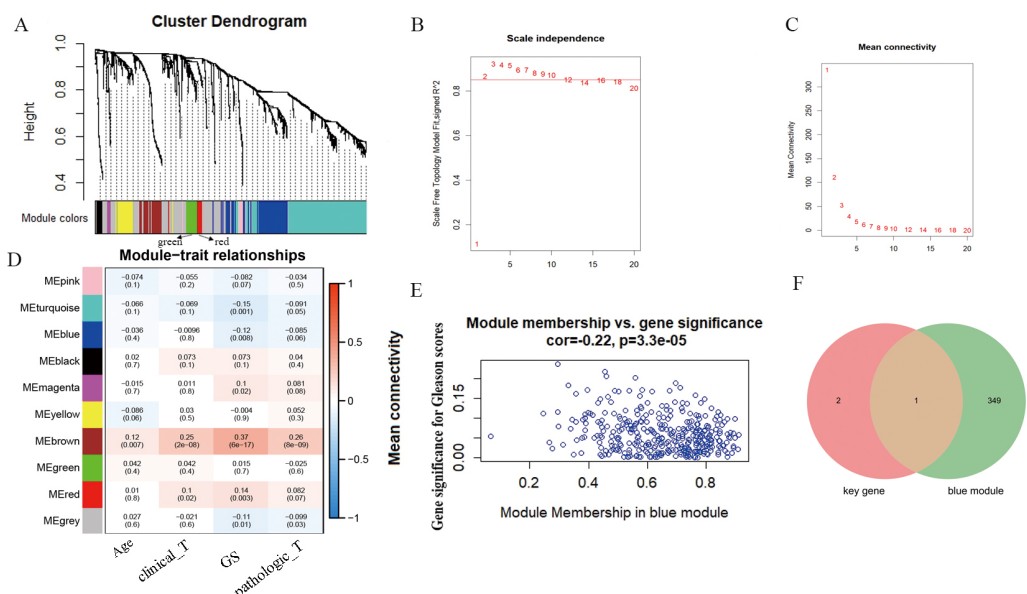

**Figure 3   WGCNA analysis and Gleason score–related gene GPX2.** (A) Hierarchical clustering tree based on the difference of adjacent values. (B and C) Topological structure analysis of soft threshold parameters. (D) Correlation between modules and clinical characteristics. The numbers represent correlation coefficients, and the numbers in parentheses represent *P* values. (E) Blue module (correlation and *P* value). (F) Venn diagrams of the overlapping DEGs between the blue module and key gene.

## WGCNA analysis and GPX2

According to WGCNA analysis and taking the correlation coefficient of 0.85 as the standard, the pickSoft threshold function was used to select the weight parameter of the adjacency matrix (soft threshold); β = 2 was the standard gene module (Figs. 3A–3C). Using the two-step method, the minimum number of genes in each gene module was set to 30, and the height of cutting branches and merging modules was set to 0.25. Finally, nine modules were obtained (Fig. 3D). Of these, we selected the blue module for this study, which was comprised of 450 genes with the correlation (r = -−0.22, *P* = 3.3e−05) (Fig. 3E). The intersection of blue module and key genes yielded only one gene GPX2 (Fig. 3F), which was used as in follow-up research.

## GPX2 expression independently predicted RFS in PCa

To further evaluate the prognostic value of GPX2 for PCa, the prognostic nomogram was constructed by integrating clinical factors and gene expression (Fig. 4D), and the correction curve was drawn to evaluate the predictive ability of the nomogram. The correction curve showed that the risks predicted by the nomogram were highly consistent with the observed RFS for 1, 3, and 5 years (Figs. 4A–4C).

## GSEA analysis

The GSEA analysis showed that GPX2 expression was mainly related to EMT and infectious disease biology, including the EMT and Wnt signaling pathways (Figs. 4E and 4F). The

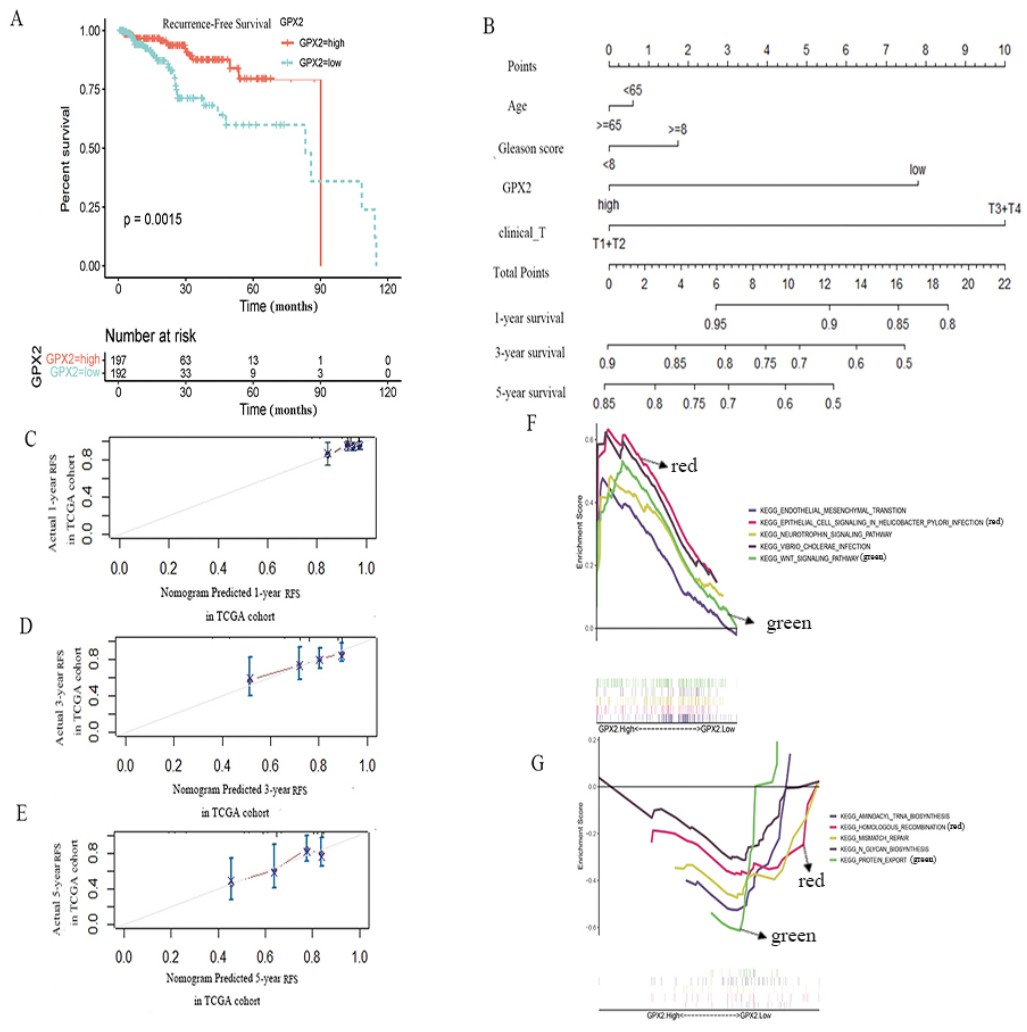

**Figure 4  GPX2 predictive PCa RFS and GSEA analysis.** (A) Subgroup analyses of RFS. (B) Postoperative prognostic nomogram for patients with PCa. The calibration curve of the nomogram for predicting RFS after 1 year (C), 3 years (D), and 5 years (E). GSEA analysis of high (F) and low GPX2 (G) expression phenotypes.

high expression of GPX2 could inhibit the activity of the aforementioned pathways, thus inhibiting the occurrence and development of PCa.

## CIBERSORT analysis

A significant difference was found in the degree of immune cell infiltration in 61 samples (20 in the GPX2 low-expression group and 41 in the GPX2 high-expression group) (Figs. 5A and 5B). Eight kinds of immune cells (activated dendritic cells, resting dendritic cells, M0 macrophages, M2 macrophages, monocytes, neutrophils, resting memory CD4 T cells, and CD8 T cells) showed GPX2 expression with significant differences (Fig. 5C).

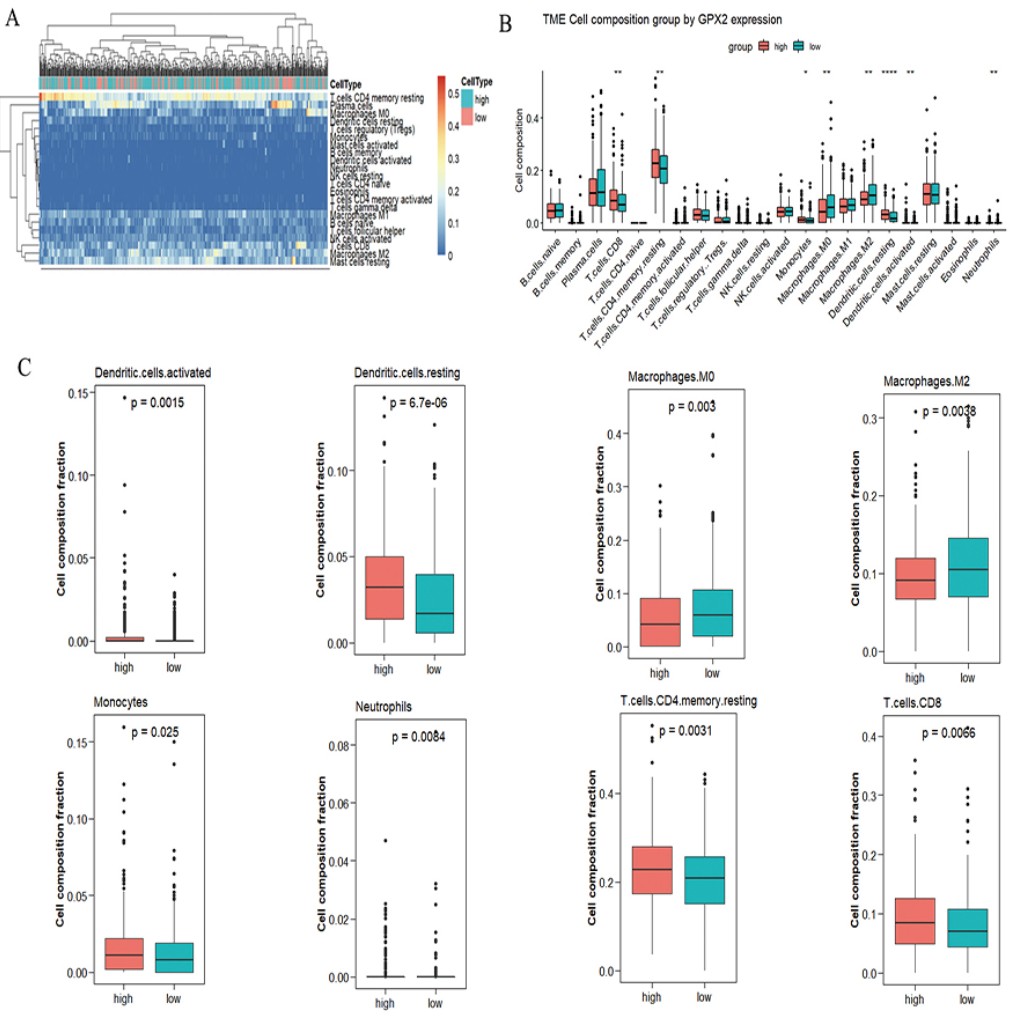

**Figure 5 GPX2 with PCa immune cells.** (A) Hot plot of immune cell infiltration in 61 PCa samples. (B) Histogram diagram of immune cell proportions in the GPX2 gene high- and low-expression groups. (C) Eight kinds of immune cells with significant GPX2 expression in PCa samples.

## GPX2 expression in PCa with the Gleason score

The immunohistochemical analysis showed that a GPX2-positive immune reaction was located in the cytoplasm and brownish yellow particles existed in the cytoplasm (Fig. 6A). GPX2 expression in benign prostatic hyperplasia tissue was significantly higher than in PCa tissue, with no significant difference in Gleason score (Fig. 6B). GSE66602 and GSE6919 datasets also confirmed the aforementioned results (Fig. 6C).

## Expression of GPX2 in PCa cells and transfection

qRT-PCR and Western blotting were used to detect the highest mRNA level of GPX2 in LNCaP and 22RV1 cells, and GPX2 was used for subsequent experiments (Fig. 6D). Western blotting showed that the expression of GPX2 in the si-GPX2 group was significantly lower than that in the si-NC group. Also, the expression of GPX2 in the over-GPX2 group

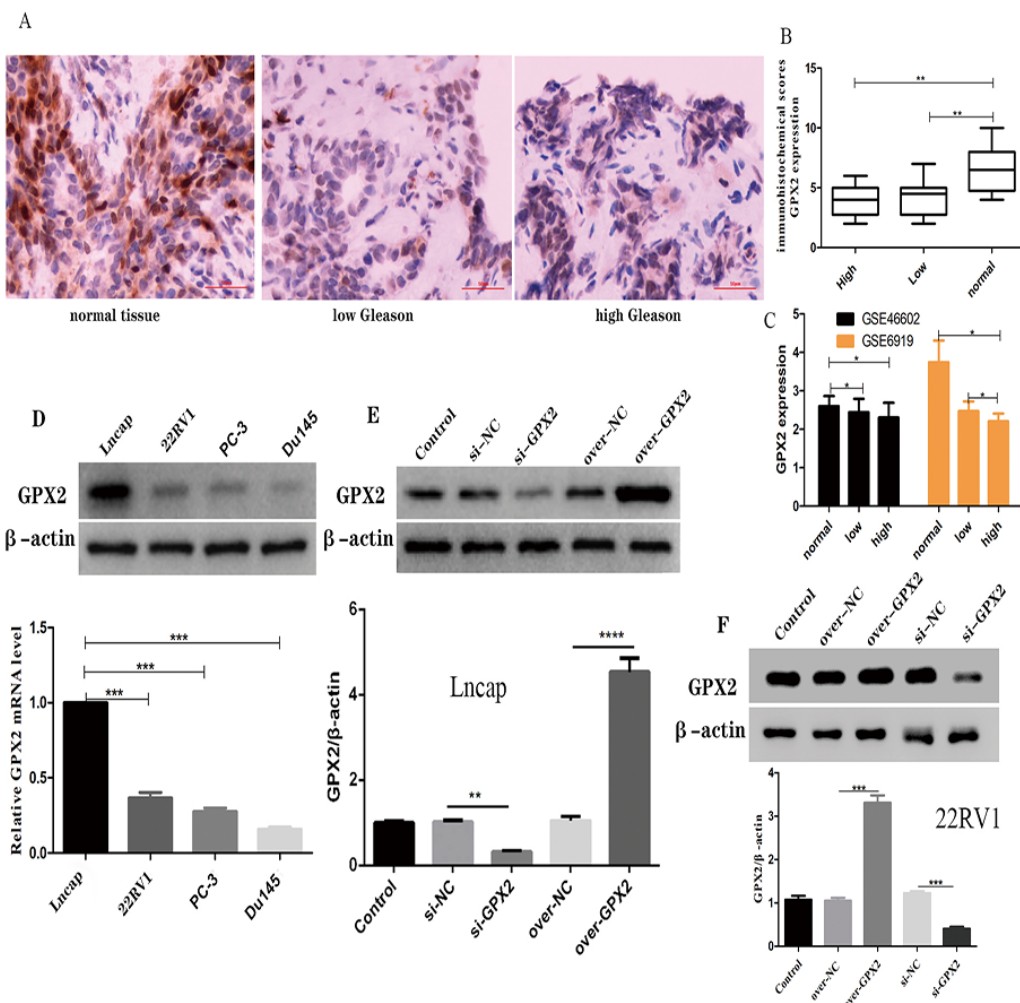

**Figure 6 Expression of GPX2 in PCa cells and transfection.** (A) Representative IHC images of GPX2 expression and Gleason score in PCa tissues and benign prostatic hyperplasia tissues. (B) IHC score of GPX2 expression (* $P < 0.5$, ** $P < 0.01$ compared with Gleason). (C) GPX2 expression in GSE66602 and GSE6919 datasets. (D) Western blotting and qRT-PCR analysis of GPX2 expression in PCa cell lines (* $P < 0.5$, ** $P < 0.01$, *** $P < 0.001$ compared with Lncap). (E and F) Western blotting of PCa cell transfection (* $P < 0.05$, ** $P < 0.01$, *** $P < 0.001$ compared with the NC group).

was significantly higher than that in the over-NC group, indicating that silencing and overexpression were successful and could be used in subsequent experiments (Figs. 6E and 6F).

## Biological behavior of GPX2 in LNCaP and 22RV1 cells

Compared with the si-NC group, the si-GPX2 group showed the inhibition of cell proliferation (Figs. 7A and 7D) and invasion (Figs. 7C and 7F), and the promotion of cell apoptosis (Figs. 7B and 7E). Compared with the over-NC group, the over-GPX2 group showed the promotion of cell proliferation (Figs. 7A and 7D) and invasion (Figs. 7C and 7F), and inhibition of apoptosis (Figs. 7B and 7E).

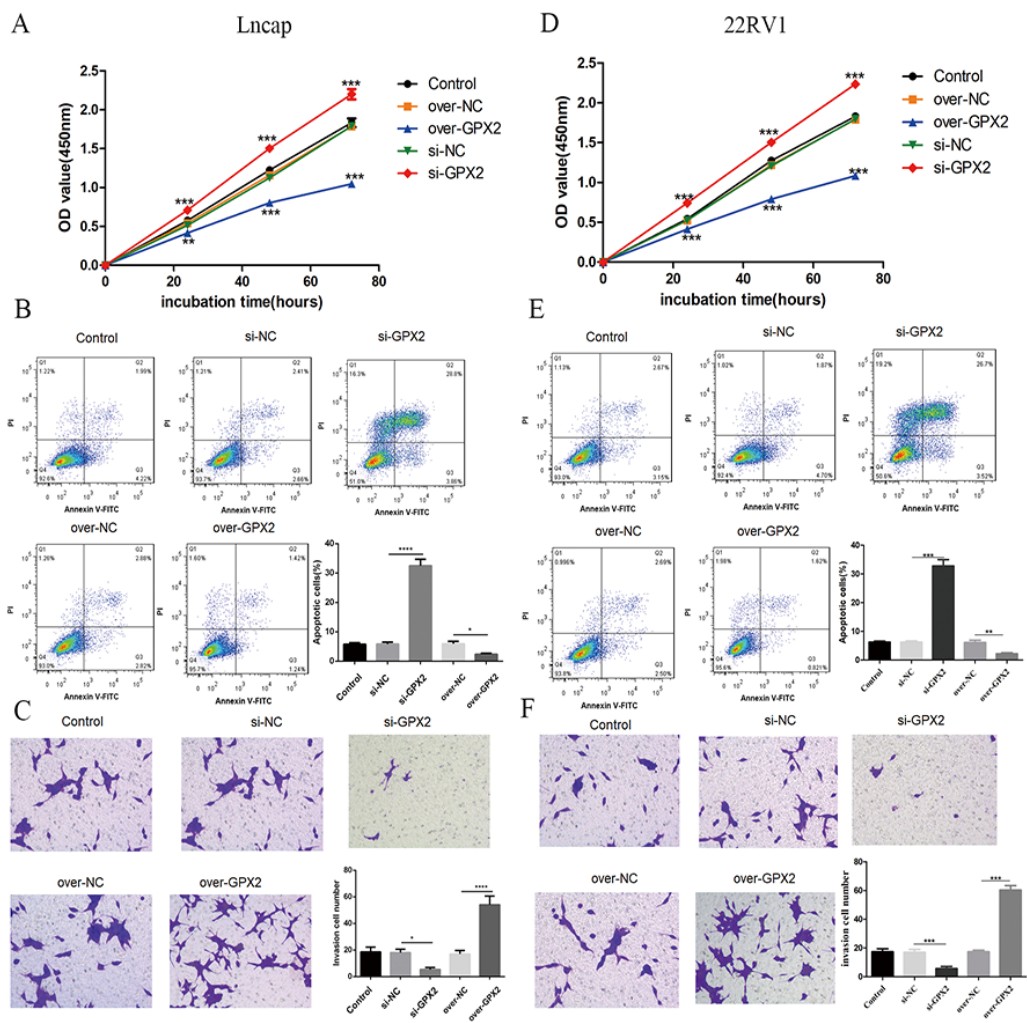

**Figure 7 Biological behavior of GPX2 in LNCaP and 22RV1 cells.** (A and D) CCK8 assay revealed that the up- and downregulation of GPX2 significantly regulated the cell viability. (B and E) Flow cytometry assay revealed that the up- and downregulation of GPX2 regulated cell apoptosis. (C and F) Transwell assay revealed that the up- and downregulation of GPX2 significantly regulated the f invasion cells (* $P <$ 0.05, ** $P < 0.01$, *** $P < 0.001$ compared with the NC group).

## GPX2 regulates the Wnt/β-catenin/EMT pathway in LNCaP and 22RV1 cells

The protein expression of Wnt3a, GSK3 β, p-GSK3 β, β-catenin, p- β-catenin, c-myc, cyclin D1, and vimentin decreased and that of E-cadherin increased in the si-GPX2 group compared with the si-NC group. The results for over-GPX2 were opposite to those for over-NC (Figs. 8A–8B and 8E– 8F). Additionally, the protein expression of β-catenin increased and that of GPX2 decreased in the si-GPX2 + SKL2001 group compared with the si-NC group (Figs. 8C–8D and 8G–8H).

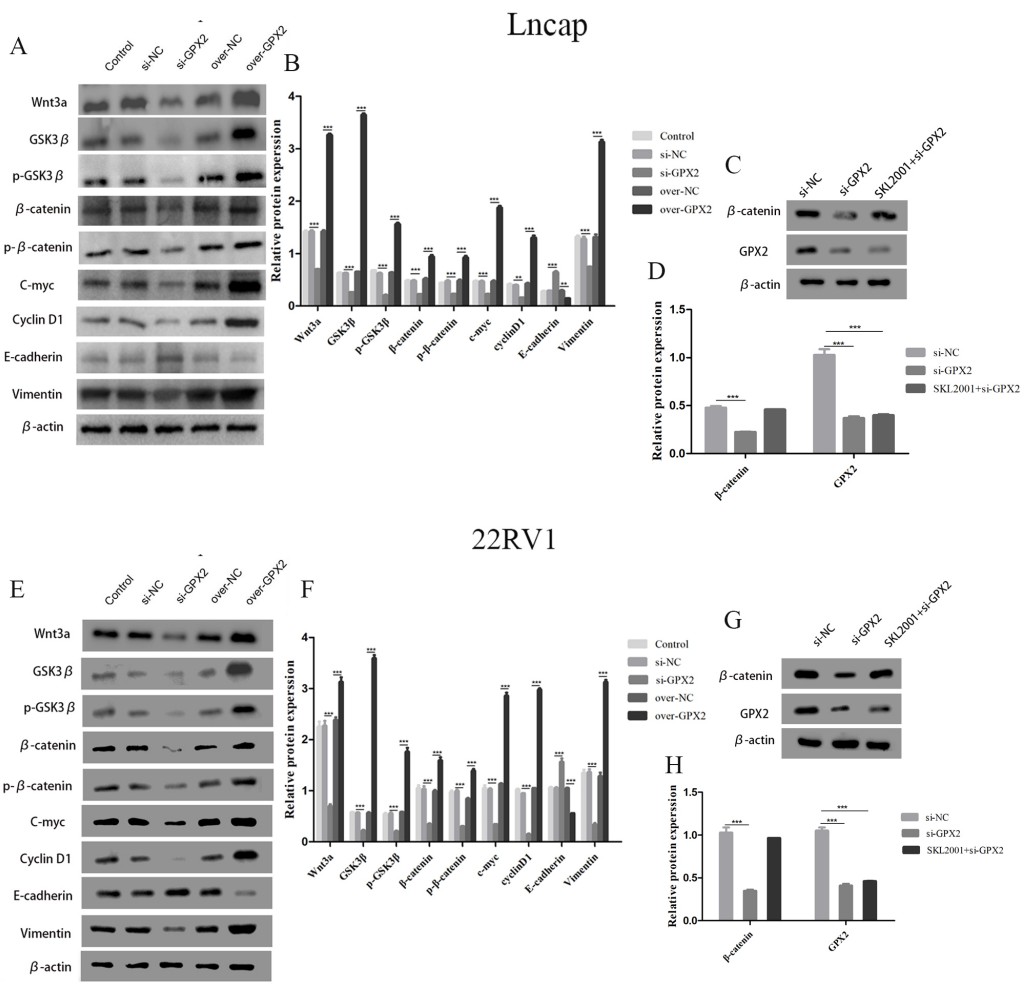

**Figure 8 GPX2 regulates the Wnt/β-catenin/EMT pathway in LNCaP and 22RV1 cells.** GPX2 regulates the Wnt/β-catenin/EMT pathway in LNCaP and 22RV1 cells. (A and B) In LNCaP cells western blot analysis revealed that the up- and downregulation of GPX2 regulated the expression of Wnt3a, GSK3 β, p-GSK3 β, β-catenin, p- β-catenin, c-myc, cyclin D1, and vimentin. (C and D) Western blot analysis of β catenin and GPX2 expression in LNCaP cells of the three groups. (E and F) In 22RV1 cells western blot analysis revealed that the up- and downregulation of GPX2 regulated the expression of β-catenin, C-myc, Cyclin D1, vimentin, and E-cadherin. (G and H) Western blot analysis of β catenin and GPX2 expression in LNCaP cells of the three groups. (* $P < 0.05$, ** $P < 0.01$, *** $P < 0.001$ compared with the NC group).

## DISCUSSION

Although the incidence rate of PCa in Asia is far lower than that of Europe and North America, the incidence and mortality rate of PCa in China has rapidly increased in recent years (*Bray et al., 2018*; *Gu et al., 2018*). The routine clinical application of prostate-specific antigen has produced good results in helping the early diagnosis of PCa (*Perera et al., 2021*). However, PCa is a clinically heterogeneous cancer with large individual differences, particularly involving the diagnosis and treatment of early tumor diagnosis and later tumor progression, tumor metastasis, and hormone resistance (*Ji et al., 2019*). Identifying the

genes related to the occurrence and development of PCa and clarifying the pathogenesis of cancer can provide a theoretical basis for preventing and treating PCa (*Giri et al., 2018*; *Velho et al., 2018*). Therefore, in-depth clinical and basic research involving a larger sample size is necessary in order to explore reliable diagnosis and treatment methods.

PCa is a complex disease affected by multiple genes. In this study, we developed a risk score based on 18 genes that was verified using GSE70768 and TCGA-PRAD datasets, with a good prediction performance. WGCNA was used to analyze the TCGA-PRAD dataset, and nine modules associated with the pathological grade, Gleason scores, TNM stage, and clinical characteristics of PCa were obtained. We selected the Gleason score and blue module for analysis and found a significant correlation (Cor = -−0.22, $P = 3.3e−05$). Then, we predicted the intersection of three parts of the model genes using the blue module, Top30, and key genes. Finally, only the core gene GPX2 was obtained. A nomogram was constructed to predict the recurrence of PCa. The nomogram could predict the possibility of recurrence in PCa patients and was more accurate than clinical indicators.

GPX2, also known as gastrointestinal-specific glutathione peroxidase, is a selenium-containing protein. It is mainly expressed in the gastrointestinal system and exerts anti-inflammatory and antioxidant effects (*Lennicke et al., 2017*). In recent years, GPX2 has been found to be highly expressed in a variety of tumors, especially inflammation-induced tumors, and may promote cell proliferation and inhibit apoptosis (*Minato et al., 2021*; *Ji et al., 2021*; *Tian et al., 2021*). GPX2 is also overexpressed in human and mouse CRPC cells and promotes the malignant proliferation of PCa cells. Inhibition of GPX2 expression significantly inhibited the proliferation of PCa cells and made them stagnate in the G2/M phase (*Naiki et al., 2014*). Inhibiting the expression of GPX2 can also improve the level of reactive oxygen species (*Wu et al., 2021*). These results showed that GPX2 had a certain correlation with tumor immunity. In this study, the high and low expression of GPX2 could influence eight kinds of immune cells to participate in the immune response of PCa. However, there have been few studies on the relationship between the expression of GPX2 and PCa prognosis and mechanism.

The Gleason score system is a PCa pathological grading system that was introduced in 1974 (*Gleason & Mellinger, 1974*). It has become the most powerful tool used to predict the prognosis of patients with PCa (*Nagpal et al., 2020*). It is closely related to the differentiation and invasion of PCa, which is of great significance for clinicians choosing and making treatment plans (*Thomsen et al., 2020*). IHC staining showed that the expression of GPX2 in PCa tissues had no significant correlation with the Gleason score; two datasets (GSE66602 and GSE6919) were used to verify the same results. Therefore, we speculated that GPX2 played an important role in PCa with no correlation to the Gleason score. We concluded that data mining must be combined with experimental verification. Furthermore, the survival prognosis of patients with high and low GPX2 expression was analyzed according to the public datasets GSE70768 and TCGA-PRAD. The results showed that the RFS time of patients in the GPX2 high-expression group was shorter than that of patients in the GPX2 low-expression group. In addition, this study used TCGA-PRAD data to construct a nomogram to predict the prognosis of patients with PCa, which helped us more intuitively understand the importance of the GPX2 expression levels in predicting PCa prognosis.

We used lentivirus transfection technology to promote the over-expression and low expression of GPX2 in LNCaP and 22RV1 cells in order to determine the role of GPX2 in the occurrence and development of PCa. The corresponding *in vitro* cell experiment results showed that inhibiting the expression of GPX2 could inhibit the proliferation and invasion of LNCaP and 22RV1 cells and induce apoptosis. Also, promoting the expression of GPX2 could promote proliferation and invasion and prevent the apoptosis of LNCaP and 22RV1 cells. The Wnt/β- catenin and EMT pathways were closely related to the occurrence and development of PCa (*Kaplan et al., 2021*; *Chaves et al., 2021*). *Nath et al. (2019)* found that Abi1 loss promoted the progression of PCa by modulating the Wnt signal and inducing EMT. *Zhang & Li (2020)* found that long noncoding RNA NORAD contributed to the metastasis of PCa *via* the Wnt/β-catenin/EMT pathway. However, the modulation of GPX2 on the Wnt/β-catenin/EMT pathway has not been reported. This study was novel in reporting that when the expression level of GPX2 changed, the proteins related to the Wnt/β-catenin/EMT pathways also changed. Therefore, we concluded that the mechanism of GPX2 in influencing the occurrence and prognosis of PCa was related to the Wnt/β-catenin/EMT signaling pathway.

However, despite the clinical significance of our findings, this study had some limitations. First, although the performance and AUC values of the calibration curve were excellent in the validation group, multicenter clinical application is still needed to further evaluate the external utility of the prognostic model. Only 262 genes were defined as genes related to the recurrence of PCa, and the construction of the prognostic model was evaluated. Some important genes might have been excluded before establishing the prognostic model. Second, GPX2 was highly expressed in benign prostatic hyperplasia compared with PCa. However, over-GPX2 promoted PCa cell proliferation and invasion and inhibited apoptosis, indicating that over-GPX2 promoted tumor progression to a certain extent. The underlying mechanism needs to be further examined.

In conclusion, the expression of GPX2 in PCa can be used as a new prognostic biomarker of RFS of PCa. GPX2 might regulate PCa progression *via* the Wnt/β-catenin/EMT pathway, and is expected to become a potential target for treating PCa.

### Funding

This study was funded by The Second Affiliated Hospital of Nanjing University of Chinese Medicine (grant number SEZ202006). The funders had no role in study design, data collection and analysis, decision to publish, or preparation of the manuscript.

### Grant Disclosures

The following grant information was disclosed by the authors:
The Second Affiliated Hospital of Nanjing University of Chinese Medicine:  SEZ202006.

### Competing Interests

The authors declare there are no competing interests.

![PeerJ]

## Author Contributions

- Ming Yang conceived and designed the experiments, performed the experiments, analyzed the data, authored or reviewed drafts of the article, and approved the final draft.
- Xudong Zhu conceived and designed the experiments, performed the experiments, prepared figures and/or tables, and approved the final draft.
- Yang Shen conceived and designed the experiments, performed the experiments, authored or reviewed drafts of the article, and approved the final draft.
- Qi He conceived and designed the experiments, performed the experiments, prepared figures and/or tables, and approved the final draft.
- Yuan Qin conceived and designed the experiments, performed the experiments, authored or reviewed drafts of the article, and approved the final draft.
- Yiqun Shao conceived and designed the experiments, performed the experiments, authored or reviewed drafts of the article, and approved the final draft.
- Lin Yuan conceived and designed the experiments, performed the experiments, authored or reviewed drafts of the article, and approved the final draft.
- Hesong Ye conceived and designed the experiments, performed the experiments, authored or reviewed drafts of the article, and approved the final draft.

## Human Ethics

The following information was supplied relating to ethical approvals (i.e., approving body and any reference numbers):

The Second Affiliated Hospital of Nanjing University of Chinese Medicine approved this study (2021SEZ-030-01).

## Data Availability

The raw data is available in the Supplemental Files and at figshare: Yang, Ming (2022): Raw dataset. figshare. Dataset. https://doi.org/10.6084/m9.figshare.20196941.v1.

## Supplemental Information

Supplemental information for this article can be found online at http://dx.doi.org/10.7717/peerj.14263#supplemental-information.

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
