# Peer review of "GPX2 predicts recurrence-free survival and triggers the Wnt/β-catenin/EMT pathway in prostate cancer"

_PeerJ, doi:10.7717/peerj.14263_

## Round 0.1 · original submission · Major Revisions

It should be noted that, as was also commented by other reviewers, a second prostate cancer cell line is needed to confirm these findings.

Reviewer 1 ·

Basic reporting

In this study, the authors proposed that GPX2 can be a biomarker being capable of predicting the recurrence of prostate cancer, and it is important in maintaining of prostate cancer cell phenotype via WNT/b-catenin signaling pathway and EMT. This study may shed some lights on the prediction of prostate cancer relapse. However, some improvements will be needed before it can be accepted for publication. Following are the comments:
1. Language editing is needed. Some description is not professional. For instance, the plot in Figure 5A is called heat map, not “hot plot”. In line 235, it should be larger sample size, instead of “larger samples”. These kinds of misleading descriptions are distributing over the whole manuscript.
2. The authors did quite a bit of analysis to predict out GPX2 as a biomarker for prostate cancer relapse and found out GPX2 could be related to WNT signaling and EMT. However, the authors did not provide solid data to show the connection either between GPX2 and WNT signaling or EMT. First, I would suggest the authors also detect if some EMT transcription factors such as Snail, Twist, Slug are changed after changes of GPX2 expression. Second, I would suggest the authors show if GSK3 and GSK3 phosphorylation are changed after modulating GPX2 level. In the meantime, I would suggest the authors use a WNT signaling activator after they reduce GPX2 or use a WNT signaling inhibitor after they increase GPX2 and see what happens to the prostate cancer cell phenotype.
3. Second prostate cancer cell line is needed to reverify the biology data.
4. Better proofreading is needed. Some typos are found. For example, in Figure 7F, one label should be oe-GPX2 or similar description showing the GPX2 is overexpressed. I see the label for this modulation in the graph is descripted as “legend”.

Experimental design

Please see details in the basic reporting.

Validity of the findings

Please see details in the basic reporting.

·

Basic reporting

Overall, the reporting and communication are adequate.

I think the publications responsible for the data in the gene expression omnibus (GEO) should be clearly cited rather than just listing the accession number (e.g. GSE46602). It would also be helpful to describe the samples included in each data set - for example 36 cases of prostate cancer and 14 normals.

The figure legends would be improved if they better described what the data was. For example, we often say "Data represent mean +/- SD of three independent experiments" or Data represent mean +/- SD of 3 replicate wells of a representative experiment"

Experimental design

The authors analyze prostate cancer recurrence in two publicly available database, narrow their search down to three genes and decide on GPX2 for further study.

Experiments to examine the potential for interaction of GPX2 with EMT genes and beta catenin signaling are a reasonable first step but are relatively superficial and limited in scope. Migration and proliferation are useful assays but aren't related to recurrence very specifically.

In vitro experiments would have been much more convincing if they included a second cell line - perhaps 22Rv1. It had higher GPX2 expression than PC3 or Du145.

Western blot analyses would have been better to also include an antibody specific to non-phosphorylated (active). beta catenin.

For analysis of clinical-variates, I encourage the authors to also examine the CAPRA-S score. (Cooperberg et al Cancer 2011 PMID: 21647869). This is the probably the most commonly used prognostic score in the field.

Validity of the findings

I was quite convinced that GPX2 is correlated with clinical prostate cancer recurrence after prostatectomy and that it is independent of at least some clinical parameters. This also appears to be a novel finding.

However, I have two main areas of concern:

1. The in vitro findings are interesting but far too preliminary to show that GPX2 acts through beta catenin and EMT to stimulate recurrence. They merely show some gene expression changes in response to GPX2 silencing or overexpression, an furthermore the change in beta catenin expression is only about two fold. The apoptosis, cell growth, and invasion assays show that GPX2 has some effect on LNCaP cells, but don't show anything about the downstream signaling.

2. The apopotosis, migration and cell growth assays have opposite changes than would be expected from the time to recurrence data (Fig 4A) and patient samples gene expression (Fig 6A). The patient data suggests that GPX2 delays recurrence but the in vitro assays show a pro-growth function which would be expected to make recurrences occur sooner.

Also, the GSEA KEGG analyses identify infectious disease biology rather than cancer immunology (vibrio cholorea group and helicobacter pylori group).

Additional comments

Overall, the findings are interesting but the conclusions are overstated. Because on the limited experimental data, I do not think that "proliferation, invasion, apoptosis, EMT or wnt-beta catenin" should be included in the title. Inclusion of these in the title leads the reader to think the support is much stronger than it is.

·

Basic reporting

the article was written in a clear English language that are easy to understand

Experimental design

the experimental design for this study is strong and they used a power tool of bioinformatics and computational analysis and validate their results in vitro

the idea of the work help to narrow the gap in understanding the relationship between the expression of GPX2 and prostate cancer progression

Validity of the findings

no comment

Additional comments

N/A

---

## Round 0.2 · Minor Revisions

As commented by the reviewer and a section editor, this manuscript did not provide direct evidence showing GPX2 regulates WNT or EMT signaling pathways. The authors need to modify the text so that they are not claiming that GPX2 is the regulator. Instead, you can indicate that the data suggest that GPX2 is involved in the signaling pathways, without absolute data, which needs to be validated with additional experiments outside the scope of the present study.

Reviewer 1 ·

Basic reporting

I am only partially satisfied with the authors’ revision. They improved language and reverified their results in one additional cell line. However, the authors did not provide direct evidence showing GPX2 regulates WNT or EMT signaling pathway. As I mentioned, the easiest way to explore that is to either use a WNT signaling activator after they reduce GPX2 or use a WNT signaling inhibitor after increasing GPX2. The authors need evidence to show that either WNT activator or inhibitor could antagonize GPX2 action after changing GPX2 expression.

Experimental design

Please see my comments in basic reporting.

Validity of the findings

No comment

·

Basic reporting

No Comment

Experimental design

No Comment

Validity of the findings

No Comment

Additional comments

I think this publication is ready for Publication

---

## Round 0.3 · Minor Revisions

Please note, as suggestions from the reviewer and section editor, the authors did not provide direct evidence showing GPX2 regulates WNT or EMT signaling pathway.

The conclusion in Abstract section “GPX2 regulated PCa progression via the Wnt/β-catenin/EMT pathway” and the last sentence in Discussion section “. GPX2 via the Wnt/β-catenin/EMT signaling pathway regulates PCa cell proliferation, invasion, and apoptosis, which is expected to become a potential target for treating PCa”. In addition, “GPX2 regulation via the Wnt/β-catenin/EMT signaling pathway in LNCaP and 22RV1 cells” in Results section and Figure 8 also need to be changed.

This paper needs to be checked carefully to avoid any possible errors before being considered for acceptation.

---

## Round 0.4 · Major Revisions

Please note (last reminder), no direct evidence showed that GPX2 could regulate PCa proliferation, invasion, and apoptosis through WNT or EMT signaling pathway.

As one reviewer suggested, “ The authors need evidence to show that either WNT activator or inhibitor could antagonize GPX2 action after changing GPX2 expression”.

Otherwise,

1)The title “GPX2 is one of three key genes predicting recurrence-free survival and regulating proliferation, invasion, and apoptosis based on Wnt/β-catenin/EMT pathway in prostate cancer” needs to be corrected.

2)In the Conclusion section of Abstract, “found that GPX2 was regulated PCa progression via the Wnt/β-catenin/EMT pathway” is better to be described as “found that GPX2 regulated PCa progression and triggered the Wnt/β-catenin/EMT pathway molecular changes”.
3)Line 296-297, “GPX2 regulates PCa progression via the Wnt/β-catenin/EMT pathway” should be written as “GPX2 might regulate PCa progression via the Wnt/β-catenin/EMT pathway”.

---

## Round 0.5 · Minor Revisions

The authors should make a revision addressing the following comment of the Section Editor,

"I don't think that the authors want to back off the claims that they have proven that GPX2 is a biomarker. They changed the wording but now it needs editing for English in addition to the problem previously: "They did not validate that GPX2 regulates PCa progression . . . they found bioinformatic evidence that suggests it may be involved, but further experimental data are needed."

In addition,

---

## Round 0.6 · accepted · Accept

This manuscript is qualified for publication.